# Impact of HPV Catch-Up Vaccination on High-Grade Cervical Lesions (CIN2+) Among Women Aged 26–30 in Northern Norway

**DOI:** 10.3390/vaccines13010096

**Published:** 2025-01-20

**Authors:** Amanda Sørensen Jørgensen, Gunnar Skov Simonsen, Sveinung Wergeland Sørbye

**Affiliations:** 1Department of Medical Biology, UiT The Arctic University of Norway, 9037 Tromsø, Norway; ajo126@uit.no (A.S.J.); gunnar.skov.simonsen@unn.no (G.S.S.); 2Department of Microbiology and Infection Control, University Hospital of North Norway, 9038 Tromsø, Norway; 3Department of Clinical Pathology, University Hospital of North Norway, 9038 Tromsø, Norway

**Keywords:** HPV vaccination, catch-up immunization, cervical intraepithelial neoplasia (CIN), HPV types 16 and 18, cervical cancer prevention, public health impact, HPV epidemiology, Norway, cervical screening programs, vaccine effectiveness

## Abstract

Background/Objectives: Human papillomavirus (HPV) is the primary cause of high-grade cervical lesions and cervical cancer worldwide. In Norway, HPV vaccination was introduced in 2009 for seventh-grade girls and extended through a catch-up program from 2016 to 2019 for women born between 1991 and 1996. This study evaluates the impact of the catch-up vaccination program on the incidence of HPV and high-grade cervical lesions in Troms and Finnmark. Methods: We analyzed data from 40,617 women aged 26 to 30 who underwent cervical screening between 2009 and 2023 in Troms and Finnmark, including 1850 women with high-grade cervical lesions (CIN2+) on biopsy. Using linear regression, we assessed trends in high-grade lesion incidence per 1000 screened women and the association between vaccination status and HPV-16/18 incidence. Results: Between 2017 and 2023, the incidence of high-grade cervical lesions significantly decreased: CIN2+ decreased by 33.4%, and CIN3+ decreased by 63.4%. Significant reductions in HPV-16/18-associated high-grade cervical lesions were observed among vaccinated women, with the proportion of CIN2+ cases due to HPV-16 and 18 decreasing from 56.8% in 2017 to 40.7% in 2023, reflecting a 55.8% reduction in the absolute number of cases caused by these high-risk HPV types. Comparing unvaccinated women aged 25–26 in 2016 and vaccinated women in 2023, HPV-16 incidence decreased from 5.1% to 0.1%, and HPV-18 incidence decreased from 3.3% to 0.0%. Conclusions: The catch-up vaccination program significantly reduced the incidence of HPV-16/18 and high-grade cervical lesions in Troms and Finnmark, even with the lower vaccination coverage observed in the catch-up program. These findings demonstrate the effectiveness of HPV vaccination programs in reducing HPV infections and associated cervical lesions.

## 1. Introduction

Human papillomavirus (HPV) is the most prevalent sexually transmitted infection worldwide, with over 200 identified types, more than 40 of which affect the genital area [1]. High-risk HPV types are implicated in nearly all cases of cervical cancer, a significant global public health challenge [2]. Despite being preventable and effectively treatable if detected early, cervical cancer remains the fourth most common cancer among women globally. In 2020, it accounted for approximately 604,127 new cases and 341,831 deaths, with an age-standardized incidence rate of 13.3 per 100,000 women-years and a mortality rate of 7.2 per 100,000 women-years [3].

In Norway, cervical cancer significantly impacts women, especially those in their 30s and 40s. Despite the well-established Cervical Screening Program, initiated in 1995, the incidence of cervical cancer did not significantly decline during the period from 2000 to 2020. In 2023, there were 325 new cases, resulting in an incidence rate of about 11.8 per 100,000 women [4]. Moreover, mortality data from 2023 showed 80 deaths, surpassing the World Health Organization’s elimination target of an incidence rate below 4 per 100,000 women [5,6].

The national screening program, which invites women aged 25–69 for regular testing, has undergone significant developments since its inception. Opportunistic Pap testing was gradually introduced in the 1970s and 1980s, but coverage was uneven, particularly lacking among women aged 40–60. With the formal establishment of national screening by the Norwegian Cancer Registry in 1995, all women within this age group were recommended to undergo Pap smear screening every three years. This screening is not mandatory but recommended, with approximately 70% coverage during the recommended three-year interval and 80% coverage over a five-year interval after two reminders. Women are required to pay a small fee [7]. The transition from undergoing a Pap smear every three years to HPV testing every five years was completed on a national level by 2023, initially relying on cytology and incorporating liquid-based cytology from 2005 to improve diagnostic accuracy and enable concurrent HPV testing. From 2015 to 2018, pilot projects facilitated the adoption of HPV testing as the primary screening method, fully implemented by 1 July 2023 [8]. This shift aims to enhance the early detection and treatment of high-grade lesions, significantly reducing both the incidence and mortality of cervical cancer [9].

Studies underscore the importance of early vaccination [10]. Research from Scotland showed that girls vaccinated at ages 12–13 had an 86% reduced risk of high-grade cervical lesions (CIN3+) compared to their unvaccinated peers, with vaccine efficacy declining as the age at vaccination increased [11]. Supporting this, Norwegian research indicates that women vaccinated before age 20 experience significantly lower risks of high-grade lesions than those vaccinated later [12].

Multiple studies confirm the preventive effects of HPV vaccination. A 2024 Norwegian study shows a significant reduction in high-grade cervical lesions (CIN2+) among women aged 20–25 years in Troms and Finnmark [13]. Data from 2022 indicate that the incidence of treatable precursors in the 1997 cohort was halved compared to the 1996 cohort, illustrating the vaccine’s impact on reducing CIN3 among women vaccinated in seventh grade [14]. Currently, vaccination coverage among girls aged 12–14 years reaches approximately 90%, with expected declines in HPV types 16 and 18 as these cohorts age into the screening brackets [15,16].

Persistent infection with high-risk HPV types is a key factor in the development of cervical cancer precursors [2]. To reduce the incidence of HPV-related diseases, Norwegian health authorities have implemented extensive vaccination measures, including a four-year catch-up program that was launched in 2016, which extends the childhood immunization schedule to include older age groups [17]. This initiative is particularly important because these cohorts continue to contribute to viral transmission. By extending vaccination to a broader segment of the population, the program aimed to accelerate the overall decline in HPV prevalence, reduce the incidence of vaccine-covered types, and thus lower the risk of cervical precancerous lesions [10,18]. In parallel, the Norwegian Cervical Screening Program employs an integrated approach in which HPV-based primary screening is followed by a cytological examination of the positive samples, thereby ensuring early detection and management of cellular abnormalities [6]. To evaluate the impact of these vaccination efforts, we analyzed the incidence of vaccine-covered HPV types, as well as the incidence of established cervical cancer precursors, in a population offered vaccination through the catch-up program. We also included an assessment of HPV samples from women within the screening age as of 2023, including those vaccinated in childhood. This approach provides a robust data foundation for assessing the long-term effects of early vaccination and offers valuable insights for further optimizing public health measures against HPV-related diseases.

The primary aim of this study is to assess the overall impact of the HPV catch-up vaccination program on the reduction of high-risk HPV types, specifically HPV-16 and HPV-18, and high-grade cervical intraepithelial neoplasia (CIN2+) in Troms and Finnmark. This analysis spans from 2009 to 2023, focusing on women aged 26–30, a critical demographic in evaluating the long-term effectiveness of the catch-up vaccination efforts. By examining the changes in HPV incidence and the incidence of CIN2+ lesions, this study aims to provide insights into the efficacy of HPV vaccinations and inform future public health strategies.

The secondary aim is to explore the broader implications of HPV type incidence changes across a wider age range, from 25 to 69 years, using data from the latest HPV DNA testing conducted in 2023 and a historical comparison from 2016. This comprehensive approach allows for an evaluation of both immediate and longitudinal impacts of HPV vaccination across different demographic groups, enhancing our understanding of vaccine efficacy in a real-world setting.

## 2. Materials and Methods

### 2.1. Data Collection

This study utilized comprehensive data from the Department of Clinical Pathology at the University Hospital of North Norway (UNN) in Troms and Finnmark, primarily sourced from the SymPathy database (Tietoevry, Espoo, Finland). This database includes records of HPV test results, cervical cytological samples, and histological findings, facilitating a robust analysis of HPV incidence and vaccine efficacy. Additionally, vaccination records from SYSVAK, Norway’s national vaccination registry, were linked to individual health data using unique personal identification numbers, enabling the precise tracking of vaccination status alongside HPV test results and histopathological outcomes.

Annually, the Department of Clinical Pathology at UNN analyzes approximately 25,000 cervical samples using the Roche Cobas 4800 HPV DNA test. This assay identifies 14 high-risk HPV types, providing specific genotype information for HPV-16 and HPV-18, while the other 12 types are reported collectively. From 2016 to 2020, co-testing with cytology and HPV mRNA was implemented to enhance screening accuracy and mitigate the incidence of cervical cancer from false-negative results. The PreTect SEE HPV mRNA test (PreTect AS, Klokkarstua, Norway) is employed to detect full-length E6/E7 transcripts of HPV types 16, 18, and 45 using nucleic acid sequence-based amplification (NASBA).

This study covers a 15-year period from 2009 to 2023, assessing the impact of the HPV vaccination program. Data analysis included HPV mRNA testing and DNA genotyping to evaluate the HPV type distribution, with vaccination coverage tracked through SYSVAK.

Norway’s HPV vaccination strategy utilizes Cervarix (2vHPV) to protect against HPV types 16 and 18, and Gardasil (4vHPV), which also covers types 6 and 11, responsible for most genital warts [19]. The vaccine type administered varied by birth cohort: girls born from 1997 to 2004 were vaccinated with Gardasil, while those born after 2005 and boys born after 2006 received Cervarix. From 2016 to 2019, a catch-up program targeted women born between 1991 and 1996, offering them 2vHPV when they were 20–25 years old [17]. Additionally, the 9vHPV vaccine (Gardasil 9), which provides broader protection against five additional HPV types, is available privately.

### 2.2. Primary Objective

The primary focus of this study is to analyze the impact of the HPV vaccination on the incidence of high-grade cervical intraepithelial neoplasia (CIN2+) and specific HPV types among women aged 26–30 years in Troms and Finnmark. This cohort participated in the National Cervical Screening Program and is analyzed based on their vaccination status and timing, reflecting exposure differences to the HPV vaccine.

### 2.3. Secondary Objective

This encompasses a broader demographic of women aged 25–69 years who participated in cervical screening using HPV DNA testing in 2023, alongside a comparative group aged 25–32 years who underwent HPV mRNA testing in 2016. This allows for a comprehensive analysis of the HPV type incidence changes over time.

### 2.4. Study Groups

Group 1 (Ages 25–26): Is composed of women born in 1997–1998, vaccinated through the childhood vaccination program in 2009–2010, and examined 13–14 years post-vaccination to assess the long-term vaccine effectiveness.

Group 2 (Ages 27–32): Includes women born in 1991–1996 who received the catch-up vaccination during 2016–2019 and were examined 4–7 years post-vaccination to evaluate the immediate impacts of the catch-up program on HPV-related cervical lesions.

Group 3 (Ages 33–69): Features women born between 1953 and 1990 who did not receive free HPV vaccination, serving as a baseline for assessing vaccine effectiveness among younger vaccinated cohorts.

### 2.5. Vaccination Coverage and Data Linkage

Coverage was high, with 61.1% of women born in 1991–1996 and 73.3% of women born in 1997–1998 vaccinated [16]. The vaccination status, linked to individual records in SYSVAK, facilitates precise comparisons of the HPV type of incidence among vaccinated and unvaccinated women, along with historical data from 2016 for unvaccinated cohorts.

### 2.6. Study Design

Primary Analysis: Repeated cross-sectional studies from 2009 to 2023 to assess the trends in CIN2+ incidence among women aged 26–30 years were undertaken.

Secondary Analysis: A descriptive cross-sectional analysis was conducted to evaluate the distribution of HPV types in all primary screening samples collected in 2023 from women aged 25–69 years, categorized into three age groups as follows: Groups 1–3. Additionally, a comparative cross-sectional analysis was performed to assess the shifts in HPV-type incidence by comparing data from groups 1 and 2 in 2023 with corresponding age groups from 2016, providing insights into changes following the implementation of HPV vaccination programs. 

### 2.7. Variables and Data Linkage

The objective of this study was to assess the impact of the HPV catch-up vaccination program on the incidence of high-grade cervical lesions (CIN2+) in women aged 26–30 years. Additionally, this study aimed to analyze the distribution of HPV types among women eligible for cervical screenings based on their vaccination status. This analysis helps to elucidate the protective effects of the HPV vaccination as it relates to variations in HPV type prevalence and the subsequent development of cervical lesions. ’Incidence’ in this context refers to the rate at which new cases of HPV infections and CIN2+ occur in the defined population during the study period. For our purposes, ’vaccinated’ is defined as having received at least one dose of a HPV vaccine. The linkage of vaccination status to individual health records in SYSVAK enabled precise comparisons of the incidence rates among vaccinated and unvaccinated women, complemented by historical data from 2016 for cohorts that were not vaccinated.

### 2.8. Statistical Methods

Descriptive statistics, including frequencies and incidence rates expressed per 1000 women, were used to calculate the incidence of HPV types and the incidence of CIN2+, CIN3+, and cervical cancer across different age and vaccination groups. Linear regression analysis was applied to investigate the trends in the incidence of CIN2+ and CIN3+ over time. The study period was divided into two segments, 2009–2017 and 2017–2023, to assess the effects before and after the introduction of the catch-up vaccination program. Statistical analyses were conducted using IBM SPSS Statistics version 29.0.1.0 (IBM Corp., Armonk, NY, USA) and R version 4.3.2 (R Foundation for Statistical Computing). A *p*-value of <0.05 was considered statistically significant.

### 2.9. Inclusion and Exclusion Criteria

The study population included 40,617 women aged 26–30 who underwent cervical screening between 2009 and 2023 in Troms and Finnmark. Among these, 1850 were diagnosed with high-grade cervical lesions (CIN2+). For women vaccinated through the catch-up program who were gradually included from 2017 to 2023, both HPV status and vaccination status were documented alongside their CIN2+ diagnoses. To avoid duplicate counts, women with multiple biopsies were included based on their first diagnosis, with the most severe CIN2+ diagnosis recorded and linked to the date of the initial diagnosis. The catch-up vaccination program began in 2017 with one cohort of women in this age group, followed by the inclusion of 2–4 new cohorts annually from 2016 to 2020. By 2021–2023, all vaccinated cohorts within the age group were represented in the study. The most severe diagnosis was recorded, along with the date of the first CIN2+ diagnosis. In 2017, one cohort of women in this age group was included in the catch-up vaccination program. From 2018 to 2020, 2–4 new cohorts were added each year, and from 2021 to 2023, all vaccinated cohorts were represented.

As a secondary objective, we investigated the HPV status in primary screening samples collected in 2023 from women aged 25–69. We included screening samples from 13,814 women who underwent HPV DNA testing as primary screening and who had no previous abnormal findings or HPV-positive tests (no symptoms, previous findings, or follow-up after conization).

To compare the incidence of HPV types with the unvaccinated cohorts from 2016, we included the results from HPV mRNA testing from 4594 women aged 25–32, originating from groups 1 and 2.

### 2.10. Ethical Approvals

This study was approved as a quality assurance project by the Regional Committee for Medical and Health Research Ethics North (REK Nord, reference number 230825). All data were anonymized, and therefore, no further approvals or informed consent were required from the participants.

## 3. Results

The Department of Clinical Pathology at UNN receives approximately 25,000 cervical cytology samples annually from women in Troms and Finnmark counties. From 2009 to 2023, a total of 40,617 women in the 26–30 age group had cervical samples in Troms and Finnmark, including 1850 women diagnosed with CIN2+, 759 with CIN3+, and 35 with cervical cancer.


**Trends in CIN2+, CIN3+, and Cervical Cancer**


The incidence rates of CIN2+, CIN3+, and cervical cancer in the 26–30 age group showed significant declines from 2017 to 2023. This decline aligns with the implementation of the HPV catch-up vaccination program, initiated in late 2016. Specifically, the rate of CIN2+ decreased from 61.3 per 1000 screened women in 2017 to 40.8 in 2023, representing a 33.4% reduction (Table 1). Linear regression analysis confirmed this decrease as statistically significant, with an estimated annual reduction of 12.1 cases per 1000 screened women (*p* < 0.01), as shown in Figure 1. Similarly, the incidence rate of CIN3+ fell from 25.7 per 1000 screened women in 2017 to 9.4 in 2023, a reduction of 63.4% (Table 1). This decline was also significant, corresponding to an estimated annual reduction of 7.8 cases per 1,000 screened women (*p* < 0.01), as presented in Figure 2. For cervical cancer, the rate decreased from 2.2 per 1000 screened women in 2017 to 0.4 in 2023, an 81.8% reduction (Table 1). However, the small number of cases resulted in this trend not reaching statistical significance (*p* > 0.05) (Figure 3).

Parallel to the reduction in disease incidence, vaccination coverage among women in this age group has increased substantially. According to Table 2, the proportion of women aged 26–30 belonging to vaccinated cohorts grew significantly after the initiation of the catch-up program in 2017. By 2021, all women in this age cohort were part of the vaccinated cohorts, with actual vaccination rates rising from 12% in 2017 to 71% in 2023. This increased vaccine uptake is correlated with the declines in CIN2+, CIN3+, and cervical cancer rates, highlighting the impact of the vaccination program on reducing the burden of HPV-related diseases in northern Norway. 


**HPV Type Distribution in CIN2+**


Following the introduction of the catch-up vaccination program, the proportion of CIN2+ cases associated with HPV types 16 and 18 has seen a significant decrease, reinforcing the impact of the vaccine on high-risk HPV infections. In 2017, there were 104 cases of CIN2+ associated with HPV-16 and 18, which constituted 56.8% of all CIN2+ cases. By 2023, this number had reduced to 46 cases, accounting for 40.7% of such cases, representing a 55.8% reduction in absolute numbers and a notable shift in the proportion of cases caused by these high-risk HPV types. This decrease was statistically significant, with an estimated annual reduction of 9.0 cases (*p* < 0.01), as illustrated in Table 3 and Figure 4. In contrast, cases associated with other HPV types decreased from 79 to 67 during the same period, a reduction of 15.2%. However, this reduction was not statistically significant (*p* = 0.6), indicating a more pronounced effect of the vaccine on HPV-16 and 18 compared to other types.


**HPV Status in Screening Tests**


In 2023, HPV DNA primary screening tests provided a comprehensive overview of the HPV infection incidence across three age groups, offering a comparison with baseline data from 2016 where available.

Group 1: In 2023, the overall HPV-positive rate was 33.3% among women aged 25–26, with 1.5% testing positive for HPV-16 and 0.7% for HPV-18. Among the vaccinated women in this group, the rates were significantly lower, at 0.1% for HPV-16 and none for HPV-18. This marks a notable reduction compared to 2016 when HPV mRNA screening showed that 5.1% of women in this age group tested positive for HPV-16 and 3.3% for HPV-18 before the implementation of the catch-up vaccination program, as detailed in Table 4.

Group 2: In 2023, women aged 27–32 had an overall HPV-positive rate of 14.7%, with 1.6% testing positive for HPV-16 and 0.4% for HPV-18. Among the vaccinated women, the rates were reduced to 0.8% for HPV-16 and 0.4% for HPV-18. In comparison, data from 2016 showed positivity rates of 2.7% for HPV-16 and 1.3% for HPV-18 in this age group prior to the catch-up vaccination program, as detailed in Table 4.

Group 3: In 2023, HPV DNA screening was also conducted for the first time in the unvaccinated age group of 33–69 years, showing HPV positivity rates of 1.0% for HPV-16 and 0.2% for HPV-18, providing baseline data for this demographic.

## 4. Discussion

Our study demonstrated a significant reduction in the incidence of high-grade cervical intraepithelial neoplasia (CIN2+ and CIN3+) among women aged 26–30 years in Troms and Finnmark from 2017 to 2023, following the introduction of the HPV catch-up vaccination program. Specifically, we observed a 33.4% reduction in CIN2+ and a 63.4% decrease in CIN3+ incidence rates over this period. These findings clearly illustrate the effectiveness of the HPV vaccination in reducing the burden of high-risk HPV types and associated cervical lesions, with a vaccination coverage of 61.1% [16]. This trend aligns with international evidence, such as data from Sweden, where girls aged 13–18 vaccinated with a coverage rate of 55% saw reductions of 46% in CIN2+ and 57% in CIN3+ [20].

We further explored the broader implications of these findings by comparing the changes in the incidence of HPV types 16/18 and the incidence of cervical lesions before and after the implementation of the catch-up vaccination program. From 2009 to 2017, before the vaccination program was fully implemented, the incidence of CIN2+ was increasing rapidly, suggesting a higher baseline rate of cervical lesions. If this trend had continued without the vaccine, we would expect about 83 cases per 1000 women screened (266 total cases) by 2023. Instead, thanks to the vaccination program, the rate dropped to about 41 cases per 1000 women screened (120 total cases), which is a reduction of around 55%. This highlights the powerful and lasting impact of the HPV vaccination, with a much larger decline than what can be seen by looking only at the years from 2017 to 2023.

Comparatively, in Denmark, where vaccination coverage reached 80%, the total reduction in the incidence of CIN3 was approximately 37% over the six years from 2013 to 2019, equating to an annual reduction of 6.5% [21]. These findings underscore the robust impact of high vaccination coverage on reducing the incidence of high-grade cervical lesions. Similar outcomes were observed in Michigan, where women vaccinated before age 20 experienced a 66% reduction in the risk of CIN3+ when they were up-to-date (UTD) on HPV vaccination [22]. The results from Italy indicated that women born in 1991–1993 who received free catch-up vaccinations at ages 18–20 had a 69% lower risk of CIN2+ compared to unvaccinated women, adjusted for age, residence, and socioeconomic status. This study also showed a lower risk for both low-grade lesions (LSIL+) and mild changes (CIN1+), as well as herd protection, where unvaccinated women in cohorts with high vaccination coverage had a reduced risk of cellular changes, indicating population protection [23].

Data from Ireland revealed a similar effect: a catch-up program with the Gardasil 4 vaccine for girls aged 17–18 years, with a coverage of 44–71.5%, resulted in a reduction in high-grade cytology, from 3.7% before vaccination (2015–2018) to 1.5% at the first screening around the age of 25 (2019–2022) [24].

A study from England showed even more dramatic results among younger cohorts: the incidence of cervical cancer was reduced by 83.9% and CIN3 by 94.3% among women vaccinated at ages 12–13, while those vaccinated at ages 14–16 saw a 71.3% reduction in cervical cancer cases. Women vaccinated at ages 16–18 experienced a more moderate effect, with a 35.5% reduction in cervical cancer cases. Overall, the vaccination program in England prevented approximately 687 cervical cancer cases and 23,192 cases of CIN3 by mid-2020, significantly reducing the incidence of cervical cancer across all socioeconomic groups, especially among the youngest vaccinated cohorts [25].

Despite the notable reductions in CIN2+ and CIN3+ in our study, the decrease in cervical cancer incidence was not statistically significant (86% reduction, *p* = 0.09). This may be attributed to the small sample sizes or the fact that it takes time for reductions in precancerous lesions to be reflected in cancer rates. A study from Scotland found that vaccination at ages 14–18 led to a significant reduction in invasive cervical cancer risk, with a cervical cancer incidence of 2.3 per 100,000 person-years in vaccinated women compared to 8.4 in unvaccinated women, showing a vaccine efficacy of 78.5% [26].

In addressing the lower HPV positivity rates observed in unvaccinated group 3, it is important to consider the comprehensive screening protocols in Norway, which historically included regular cervical cytology (Pap smear) screening at ages 25, 28, and 31. This rigorous screening likely led to the early detection and treatment of high-grade cervical lesions (CIN2+), reducing the prevalence of high-risk HPV infections by the time women undergo their first HPV DNA test at age 34 [6,27]. Since 2023, this protocol has transitioned to HPV testing as the primary screening method for women aged 25–33 years, enhancing the detection of HPV infections even earlier [6,28]. The historical cohort effect, where older generations were less exposed to high-risk HPV types before the widespread dissemination of HPV, coupled with the natural decline in HPV infection due to age-related immune clearance, further contributes to the observed lower positivity rates in this group [29,30]. Additionally, behavioral factors such as changes in sexual behavior over time and across different age groups might influence these findings [31]. Notably, even though the bivalent HPV vaccine reduces the incidence of HPV types 16 and 18, many young women still test positive for other HPV types not covered by the vaccine, suggesting ongoing exposure and transmission dynamics that may vary by age group [32]. These factors together explain the variance in HPV positivity rates across different demographic groups, underscoring the complex interplay of vaccination, screening, and behavioral patterns in shaping the epidemiology of HPV in the population.

Our findings demonstrate significant reductions in high-grade cervical lesions, underscoring the importance of high vaccination coverage and early administration of the HPV vaccine for effective protection against HPV-related diseases [16]. Although we observed effects on precancerous lesions within a few years from vaccination, the nature of cervical cancer development suggests that a full assessment of the vaccine’s impact on cervical cancer incidence may require a longer follow-up period. The long-term benefits, particularly in reducing cervical cancer, are expected to manifest after a follow-up of at least 10–15 years, given the prolonged progression time from HPV infection to cancer [33].

The linear regression analysis based on data from the Cancer Registry shows a statistically significant decrease in cervical cancer incidence among women aged 25–29 years (*p* < 0.05) and 35–39 years (*p* < 0.05) from 2017 to 2023, suggesting a real reduction in these age groups [34].

In the period following the catch-up vaccination program (2017–2023), we observed a statistically significant 56% reduction in the incidence of HPV-16/18 infections among vaccinated women. While our study confirmed a significant association between vaccination status and a reduction in HPV-16/18 infections, no significant decrease was observed for other HPV types.

The strong effect against HPV-16/18, combined with the limited impact on other HPV types, may be attributed to the choice of the bivalent Cervarix vaccine in the catch-up program. This vaccine primarily targets HPV types 16 and 18, which are responsible for at least 70% of cervical cancer cases but less than 50% of all CIN2+ cases. Additionally, 36.8% of CIN2+ cases are linked to HPV types 31, 33, 45, 52, and 58, which are covered by the Gardasil 9 vaccine [19,32]. While Cervarix offers some cross-protection against HPV-31, 33, and 45, this effect may decrease over time, and its long-term durability is uncertain [35,36]. As such, we cannot expect the same reduction in CIN2+ for these HPV types in women vaccinated with Cervarix compared to those vaccinated with Gardasil 9.

These findings underscore the need for vaccines with broader coverage to provide more comprehensive protection against HPV-related diseases. By including additional high-risk HPV types in the vaccine, it would be possible to further reduce the incidence of high-grade cervical lesions.

Vaccinated individuals demonstrate significantly lower rates of infection with multiple HPV types compared to their unvaccinated counterparts. This is particularly evident in younger cohorts, where vaccinated girls aged 17 and 21 years show markedly lower incidence of concurrent HPV infections, highlighting the vaccine’s effectiveness against multiple strains [37,38].

Gardasil 9, in particular, has proven highly effective, reducing the incidence of CIN from high-risk HPV types by over 96% and significantly decreasing the need for invasive procedures such as biopsies and conizations [39]. The long-term superiority of the nine-valent HPV vaccine (9vHPV) over the quadrivalent vaccine (qHPV) is evident, with studies showing a dramatic reduction in high-grade cervical disease incidences in those receiving the 9vHPV, underscoring its enhanced protective effect against additional HPV types [40].

In Norway, vaccination has also led to significant public health benefits beyond cervical disease prevention. The quadrivalent vaccine has shown effectiveness against genital warts, especially in younger vaccinated populations, with a decrease in efficacy noted as the age at vaccination increases [36,41]. Transitioning to Gardasil 9 could prevent a substantial number of HPV-related cancer cases and other conditions, such as genital warts and respiratory papillomatosis, offering considerable cost savings and health benefits over time [42].

A long-term follow-up across Denmark, Sweden, and Norway reveals that the nonavalent vaccine provides 100% efficacy against CIN2+ lesions from the covered HPV types in vaccinated HPV-negative women, confirming the broad-spectrum benefits of early vaccination and its potential to further reduce HPV-related diseases [43,44].

Comparing the HPV incidence in vaccinated versus unvaccinated women aged 25–26 from 2016, we observed that HPV-16 incidence decreased from 5.1% to 0.1%, and HPV-18 incidence reduced from 3.3% to 0.0%, following the childhood vaccination program, which achieved an average coverage of 73.3% [16]. A 2018 study found similar reductions: five years post-vaccination, 17-year-olds born in 1997 showed a 75% reduction in HPV-16 and an 86% reduction in HPV-18, compared to unvaccinated 17-year-olds born in 1994 [17].

## 5. Strengths and Limitations

### 5.1. Strengths

Our study utilizes the comprehensive SymPathy database, which includes all HPV analyses, cytology, and tissue samples from Troms and Finnmark, ensuring robust data reliability for our analysis. The use of HPV mRNA tests over a 15-year observation period allows for a detailed, long-term trend analysis, capturing the sustained effects of vaccination strategies. Diagnostic accuracy is further enhanced by dual pathologist reviews and the use of adjunctive p16 immunostaining at the University Hospital of North Norway (UNN). Advanced digital tools such as EagleEye AI were employed to minimize diagnostic variability, ensuring consistency in the results. These elements, combined with the real-world clinical setting of our data, lend practical relevance and applicability to our findings concerning the effectiveness of HPV vaccines.

### 5.2. Limitations

While our study provides valuable insights into the impact of HPV vaccination in Troms and Finnmark, the geographical specificity of the data may limit its applicability to other regions with differing health behaviors or vaccination coverage. Furthermore, this study only included women who participated in the screening program, introducing potential selection bias that might affect the representativeness of the findings. This bias could skew estimates of HPV infection and CIN2+ prevalence, reducing the generalizability of the results. Another inherent limitation of our approach was that HPV status and vaccination status analyses were conducted only among women with CIN2+ or those screened for HPV types 16 and 18. Although our study encompasses all screened women in Troms and Finnmark, individual HPV vaccination statuses were not available for women who did not attend a screening, those without CIN2+, or those not infected with HPV types 16 or 18. Despite this, given the high coverage rates of the HPV vaccine in vaccinated cohorts and the minimal opportunistic vaccinations outside the national program, we believe our findings are representative of the screened population in Norway. Additionally, we acknowledge that uncontrolled confounding factors such as variations in sexual behavior, smoking habits, and contraceptive use over the study period could have influenced the observed effects of HPV vaccination, thereby complicating the isolation of its direct impacts on the study outcomes. Finally, regarding the ultimate endpoint of cervical cancer incidence, the current follow-up duration and the statistical power of this study are insufficient to yield significant findings on the reduction of cervical cancer burden. This will remain a critical focus as more longitudinal data become available, allowing for a more comprehensive assessment of the long-term impact of HPV vaccination programs.

## 6. Conclusions

This study suggests that the reduction in HPV-16/18 incidence among vaccinated women is associated with a decrease in high-grade cervical lesions, providing evidence supporting the effectiveness of HPV vaccination, even with the lower vaccination coverage observed in the catch-up program. Our findings also indicate that both the choice of vaccine and the timing of vaccination may significantly influence the incidence of HPV infections and related diseases. By drawing on the experiences of other countries and refining national strategies, Norway may be able to accelerate its progress in reducing HPV-related diseases and move closer to the goal of eliminating cervical cancer. Continued commitment to vaccination, screening, and treatment is essential for protecting future generations from HPV and HPV-related cancers.

## Figures and Tables

**Figure 1 vaccines-13-00096-f001:**
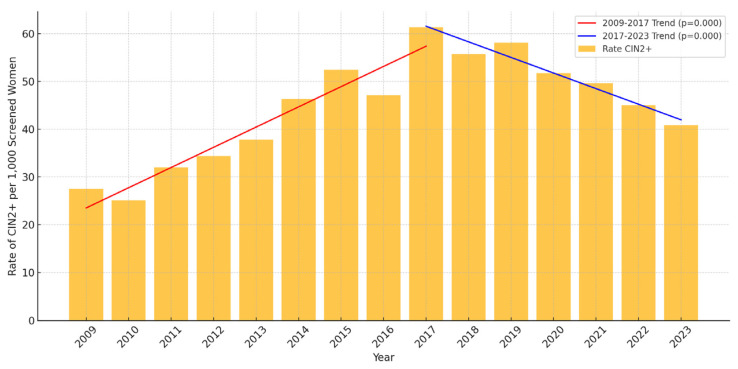
Incidence of CIN2+ per 1000 screened women aged 26–30 in Troms and Finnmark (2009–2023). Linear regression analysis demonstrates statistically significant trends with *p* < 0.01 for both periods, 2009–2017 and 2017–2023. For details on the proportions of women belonging to vaccinated cohorts during these years, please refer to Table 2.

**Figure 2 vaccines-13-00096-f002:**
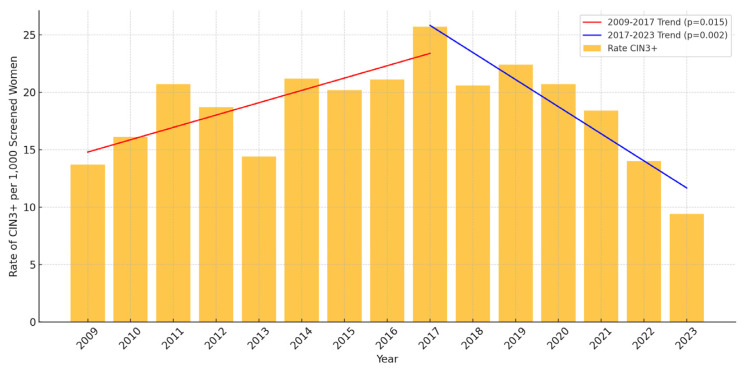
Incidence of CIN3+ per 1000 screened women aged 26–30 in Troms and Finnmark (2009–2023). Linear regression analysis demonstrates statistically significant trends with *p* < 0.05 for the period 2009–2017 and *p* < 0.01 for 2017–2023. For details on the proportions of women belonging to vaccinated cohorts during these years, please refer to Table 2.

**Figure 3 vaccines-13-00096-f003:**
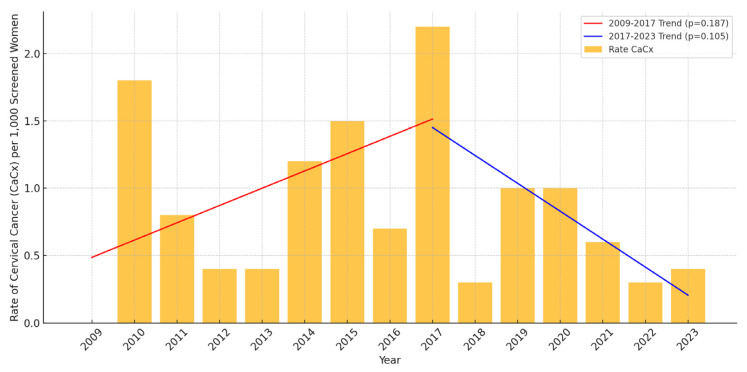
Incidence of cervical cancer per 1000 screened women aged 26–30 in Troms and Finnmark (2009–2023). Linear regression analysis demonstrates no statistically significant trends, with *p* > 0.05 for both the periods 2009–2017 and 2017–2023. For details on the proportions of women belonging to vaccinated cohorts during these years, please refer to Table 2.

**Figure 4 vaccines-13-00096-f004:**
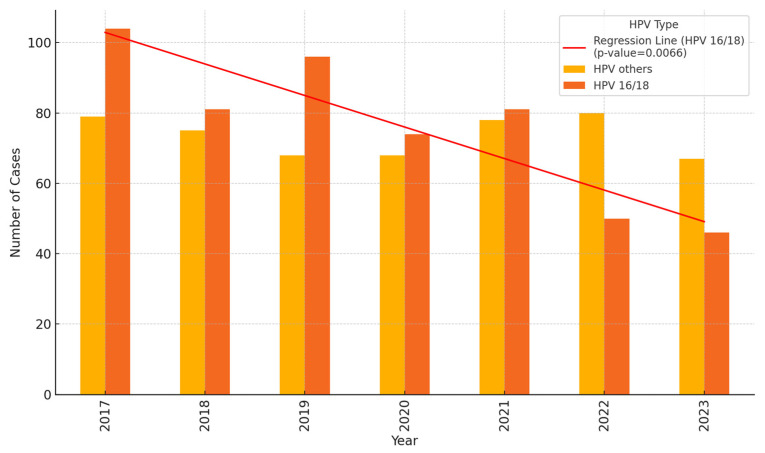
Incidence of HPV-16/18 and other HPV types from 2017–2023 among women with CIN2+ aged 26–30 years in Troms and Finnmark. For details on the proportions of women belonging to vaccinated cohorts during these years, please refer to Table 2.

**Table 1 vaccines-13-00096-t001:** Women 26–30 years of age screened with cervical cytology or HPV tests in 2009–2023 and the number of CIN2+, CIN3+, and cancer.

Year	Screening Tests	CIN2+	Rate	CIN3+	Rate	Cancer	Rate
2009	2111	58	27.5	29	13.7	0	0.0
2010	2232	56	25.1	36	16.1	4	1.8
2011	2372	76	32.0	49	20.7	2	0.8
2012	2294	79	34.4	43	18.7	1	0.4
2013	2435	92	37.8	35	14.4	1	0.4
2014	2550	118	46.3	54	21.2	3	1.2
2015	2672	140	52.4	54	20.2	4	1.5
2016	2843	134	47.1	60	21.1	2	0.7
2017	3228	198	61.3	83	25.7	7	2.2
2018	3016	168	55.7	62	20.6	1	0.3
2019	2996	174	58.1	67	22.4	3	1.0
2020	2902	150	51.7	60	20.7	3	1.0
2021	3266	162	49.6	60	18.4	2	0.6
2022	2933	132	45.0	41	14.0	1	0.3
2023	2767	113	40.8	26	9.4	1	0.4

Screening Tests = number of women screened, either with cervical cytology (Pap test) or HPV test. CIN2+ = CIN2, CIN3, ACIS, and cervical cancer; CIN3+ = CIN3, ACIS, and cervical cancer. Rate = number of cases per 1000 screened women.

**Table 2 vaccines-13-00096-t002:** Birth years of women aged 26–30 during the period 2009–2023, proportions of women belonging to vaccinated cohorts, and the percentages of women actually vaccinated.

Year	Birth Year Range	% in Age Cohort	% Actually Vaccinated
2009	1979–1983	0.0	0.0
2010	1980–1984	0.0	0.0
2011	1981–1985	0.0	0.0
2012	1982–1986	0.0	0.0
2013	1983–1987	0.0	0.0
2014	1984–1988	0.0	0.0
2015	1985–1989	0.0	0.0
2016	1986–1990	0.0	0.0
2017	1987–1991	20.0	12.0
2018	1988–1992	40.0	24.0
2019	1989–1993	60.0	35.0
2020	1990–1994	80.0	44.0
2021	1991–1995	100.0	56.0
2022	1992–1996	100.0	62.0
2023	1993–1997	100.0	71.0

Age Cohort = Proportions of women belonging to vaccinated cohorts.

**Table 3 vaccines-13-00096-t003:** Women 26–30 years of age with CIN2+ in 2017–2023 with HPV-16/18 versus other HPV types.

Year	HPV-16/18	HPV Others	HPV-16/18 (%)
2017	104	79	56.8
2018	81	75	51.9
2019	96	68	58.5
2020	74	68	52.1
2021	81	78	50.9
2022	50	80	38.5
2023	46	67	40.7

**Table 4 vaccines-13-00096-t004:** HPV genotype distribution by age group, year, and vaccination status.

Age Group	Year	Status	HPV-16	HPV-18	Number Screened
25–26	2016	Unvaccinated	5.1%	3.3%	1381
25–26	2023	Vaccinated	0.1%	0.0%	844
27–32	2016	Unvaccinated	2.7%	1.3%	3212
27–32	2023	Vaccinated	0.8%	0.4%	1894

## Data Availability

The raw data supporting the conclusions of this article will be made available by the authors upon request.

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
