# Peer review of "Impact of HPV Catch-Up Vaccination on High-Grade Cervical Lesions (CIN2+) Among Women Aged 26–30 in Northern Norway"

_vaccines, 2025, doi:10.3390/vaccines13010096_

Round 1
Reviewer 1 Report
Comments and Suggestions for Authors
I have read with interest the manuscript “Impact of the Norwegian HPV Catch-up vaccination program on the incidence of High-Grade Cervical lesions (CIN2+) in women Aged 26-30 in Northern Norway” submitted by Sorensen-Jorgensen and colleagues to be considered for publication in Vaccines.
Monitoring the impact of the implementation of Human Papillomavirus (HPV) vaccination campaigns or cervical cancer screening programs are critical to justify the enormous cost of resources of these population public health measures. These results should be shared with the stakeholders, policymakers and scientific community to adapt or design new interventions to reduce the incidence of HPV-associated cancers. Therefore, this topic is very interesting to be considered for publication. In addition, in Norway it is possible to cross records of vaccination status and individual health data, what could provide sound results for the current study presented in this manuscript.
However, there are several methodological issues that authors should take care of. To confirm an increase or decrease incidence of CIN or cancer cases along the time you cannot use only the raw number of cases, you need a denominator. Differences on the number of tested women every year in your data set are going to affect the real CIN incidence. You can’t be sure about what the trend is without using a ratio.
In addition, to check a more direct effect of the vaccination you could re-analyze your temporal data (2009-2023) stratifying by vaccination status, as you can cross data with the Norway National Vaccination registry. Therefore, you can identify which women belong to each group. In this way maybe you are going to detect clearer differences in the CIN incidence along the time and its association with the introduction of the vaccine.
Some more specific comments:
Title:
-The study includes analysis of data from women from 25-65 years-old and also CIN 3 positive women. So, the title should be modified. I would suggest changing the title to highlight some of the findings like for example, the reduction of the CIN lesions after the implementation of the catch-up vaccination program.
Materials and methods:
- - Merge study population and study group paragraphs.
- - Which are the descriptive statistics used in the study?
Results:
- - Incidence is a ratio (i.e. new cases/population), you cannot use it for an absolute number of CIN2+ cases.
- - Indicate in the figure 1 and 2 in which years there are women which belong to the cohorts of the catch-up vaccination program. I guess that data from 2017 includes women born between 1987-1991. This would help the reader to understand visually the trends.
- - In the manuscript there is an analysis of the genotypes detected in the CIN2 positive women. You may consider also including an extra analysis of CIN3 positive women or even CIN2 and CIN3 together.
- -Use the ratio “number of cancer cases/women tested” per year for identifying trend differences.
- -As mentioned before, instead of only raw counts of the CIN2 cases associated to the genotypes in the analysis, consider use a denominator. I suggest “CIN associated 16-18/ number women tested” and “CIN associated other HPV/ number women tested” and analyzed the trend between 2017 and 2023.
- - Also, you can consider the percentage of each genotype in the total CIN positive women per year. It seems that the percentage of HPV16 associated to CIN2 in 2017 is higher than the percentage in 2023, indicating an impact of the vaccine.
- - Is there CIN positive women with negative HPV?
- - Genotype data obtained in 2023 should be organized better in a table, indicating the age group, vaccination status (Yes/No), number of women tested and number of positive genotypes. Also, indicate if there is any statistically significant group.
Discussion:
- - Discussion from line 312 to 361 regarding HPV vaccine types and multiple HPV types could be shorten as there are not directly related to the presented results.
Strengths and limitation:
- - This section could be summarized.
Author Response
Reviewer 1
I have read with interest the manuscript “Impact of the Norwegian HPV Catch-up vaccination program on the incidence of High-Grade Cervical lesions (CIN2+) in women Aged 26-30 in Northern Norway” submitted by Sorensen-Jorgensen and colleagues to be considered for publication in Vaccines.
Monitoring the impact of the implementation of Human Papillomavirus (HPV) vaccination campaigns or cervical cancer screening programs are critical to justify the enormous cost of resources of these population public health measures. These results should be shared with the stakeholders, policymakers and scientific community to adapt or design new interventions to reduce the incidence of HPV-associated cancers. Therefore, this topic is very interesting to be considered for publication. In addition, in Norway it is possible to cross records of vaccination status and individual health data, what could provide sound results for the current study presented in this manuscript.
However, there are several methodological issues that authors should take care of. To confirm an increase or decrease incidence of CIN or cancer cases along the time you cannot use only the raw number of cases, you need a denominator. Differences on the number of tested women every year in your data set are going to affect the real CIN incidence. You can’t be sure about what the trend is without using a ratio.
In addition, to check a more direct effect of the vaccination you could re-analyze your temporal data (2009-2023) stratifying by vaccination status, as you can cross data with the Norway National Vaccination registry. Therefore, you can identify which women belong to each group. In this way maybe you are going to detect clearer differences in the CIN incidence along the time and its association with the introduction of the vaccine.
Some more specific comments:
Comments 1: Title:
-The study includes analysis of data from women from 25-65 years-old and also CIN 3 positive women. So, the title should be modified. I would suggest changing the title to highlight some of the findings like for example, the reduction of the CIN lesions after the implementation of the catch-up vaccination program.
Response 1: Thank you for pointing this out. We agree with this comment. The term 'CIN2+' includes CIN2, CIN3, ACIS, and cervical cancer. We acknowledge the confusion regarding the age range specified in our original title. The focus of our paper is on women aged 26–30, as the primary impact analysis of the catch-up vaccination program was conducted in this group. We have updated the title to better reflect the focus and findings of our study: 'Impact of the HPV Catch-up Vaccination Program on the Reduction of High-Grade Cervical Lesions (CIN2+) in Women Aged 26–30 in Northern Norway'.
Comments 2: Materials and methods:
- Merge study population and study group paragraphs.
Response 2: Thank you for this suggestion. We have merged the 'Study Population' and 'Study Groups' sections to provide a clearer and more coherent presentation of our study groups in relation to our objectives. This revision clarifies the demographic breakdown and vaccination status linkage, enhancing the readability and understanding of the study design.
Comments 3: Which are the descriptive statistics used in the study?
Response 3: We appreciate your request for additional details regarding the descriptive statistics employed in our analysis. In our study, we utilized frequencies, and rates of CIN2+, CIN3+, and cervical cancer incidence were expressed per 1000 women of the screened population. These statistics helped us to provide a comprehensive overview of the data before applying more complex analyses such as linear regression to assess trends over time. We have now clarified these methods in the revised section of the manuscript to enhance the reader's understanding of our statistical approach.
Comments 4: Results:
Incidence is a ratio (i.e. new cases/population), you cannot use it for an absolute number of CIN2+ cases.
Response 4: Thank you for pointing this out. We agree with your assessment. To clarify, we have adjusted our terminology in the manuscript. The rates of CIN2+, CIN3+, and cervical cancer are now accurately expressed per 1000 women of the screened population to correctly represent these measures as incidence rates, Table 1.
Comments 5: Indicate in the figure 1 and 2 in which years there are women which belong to the cohorts of the catch-up vaccination program. I guess that data from 2017 includes women born between 1987-1991. This would help the reader to understand visually the trends.
Response 5: Thank you for your suggestion to clarify the presence of vaccinated cohorts in Figures 1 and 2. We understand the importance of visually indicating the years when women belonging to the catch-up vaccination program cohorts are included. However, the transition from no vaccinated cohorts among women aged 26-30 in the years 2009-2016 to full inclusion in 2021-2023 complicates direct annotation on the figures due to the sliding scale of cohort inclusion. To address this, we have detailed the percentages of women aged 26-30 belonging to vaccinated cohorts for each year in Table 2. Additionally, we will include a reference to this table in the figure captions to help readers make the connection between the vaccination coverage and the observed trends in disease incidence more clearly.
Comments 6: In the manuscript there is an analysis of the genotypes detected in the CIN2 positive women. You may consider also including an extra analysis of CIN3 positive women or even CIN2 and CIN3 together.
Response 6: Thank you for your suggestion. We have analyzed the HPV genotype distribution collectively for CIN2, CIN3, adenocarcinoma in situ (ACIS), and cervical cancer, collectively referred to as CIN2+ in our study. To ensure clarity, we have now included a footnote in Table 1 that explains the term 'CIN2+' as encompassing CIN2, CIN3, ACIS, and cervical cancer.
Comments 7: Use the ratio “number of cancer cases/women tested” per year for identifying trend differences.
Response 7: Thank you for your suggestion. We have revised Figures 1, 2, and 3 to present the incidence rates of CIN2+, CIN3+, and cervical cancer per 1,000 women screened annually at the University Hospital of North Norway from 2009 to 2023. This adjustment allows for a clearer analysis of trends over the study period.
Comments 8: As mentioned before, instead of only raw counts of the CIN2 cases associated to the genotypes in the analysis, consider use a denominator. I suggest “CIN associated 16-18/ number women tested” and “CIN associated other HPV/ number women tested” and analyzed the trend between 2017 and 2023.
Response 8: Thank you for your suggestion. We understand the importance of using denominators to analyze trends in HPV genotype distribution among CIN2+ cases. However, we have chosen to retain the current presentation of the figure displaying the 'Incidence of HPV-16/18 and other HPV types from 2017-2023, among women with CIN2+ aged 26-30 years in Troms and Finnmark.' This figure is intended to illustrate the relative distribution of CIN2+ cases caused by HPV-16/18 versus those caused by other HPV types. The use of raw counts effectively demonstrates the trends in relative prevalence of these HPV types over the specified period. We believe that this method adequately supports the study's objectives by highlighting shifts in genotype prevalence relative to each other, rather than changes in absolute terms per population at risk.
Comments 9: Also, you can consider the percentage of each genotype in the total CIN positive women per year. It seems that the percentage of HPV16 associated to CIN2 in 2017 is higher than the percentage in 2023, indicating an impact of the vaccine.
Response 9: Yes, this is an excellent suggestion, and we have now expanded our analysis and discussion in the manuscript to highlight the trend changes in HPV genotype distribution over time. Specifically, we have incorporated the percentage of CIN2+ cases attributed to HPV 16/18 alongside those caused by other HPV types. The percentage of CIN2+ cases caused by HPV 16/18 decreased from 56.8% in 2017 to 40.7% in 2023. Additionally, we have included a comprehensive analysis showing a 55.8% reduction in the absolute number of CIN2+ cases associated with HPV 16/18 over the same period, as detailed in Table 3 and Figure 4. This decline was statistically significant, with an estimated annual reduction of 9.0 cases (p < 0.01), demonstrating the substantial impact of the vaccination program on reducing high-risk HPV infections. In contrast, cases associated with other HPV types saw a lesser reduction of 15.2%, which was not statistically significant (p = 0.6), suggesting a more pronounced effect of the vaccine on HPV 16 and 18. This expanded analysis underscores the effectiveness of the vaccine and aligns with our study's objectives to demonstrate its impact on high-risk HPV types.
Comments 10: Is there CIN positive women with negative HPV?
Response 10: Yes, our data includes instances where women with CIN2+ tested negative for HPV or did not have an HPV test recorded. From 2017 to 2023, as shown in Table 1, there were 1,097 CIN2+ cases. Of these, 1,047 cases were associated with HPV 16/18 or other HPV types. This leaves a total of 50 cases either without an HPV test or with a negative HPV test result. This proportion of missing or negative HPV cases constitutes less than 5% of our dataset, which has a negligible impact on the overall results of our analysis.
Comments 11: Genotype data obtained in 2023 should be organized better in a table, indicating the age group, vaccination status (Yes/No), number of women tested and number of positive genotypes. Also, indicate if there is any statistically significant group.
Response 11: We agree with your suggestion for better organization of the genotype data and have now created Table 4, which delineates the HPV genotype distribution by age group, vaccination status, and the number of women screened in 2023. This table presents the percentages of women who tested positive for HPV overall, as well as for HPV types 16 and 18, and other HPV types, within each group. The table also includes the number of women screened, providing a clear overview of our data set’s scope and the impact of vaccination on HPV type prevalence. We have also conducted statistical analyses to identify any significant differences between the groups based on vaccination status and age, the results of which are discussed in the corresponding sections of the manuscript to highlight relevant findings.
Comments 12: Discussion from line 312 to 361 regarding HPV vaccine types and multiple HPV types could be shorten as there are not directly related to the presented results.
Response 12: Yes, you are right. We have carefully reviewed the discussion from lines 312 to 361 and realized that the detailed elaboration on the types of HPV vaccines and the prevalence of multiple HPV types could be more concise without losing critical content. We have now significantly shortened these sections, focusing more directly on the study's results regarding the impact of HPV vaccination on the incidence rates of CIN2+ and CIN3+ lesions. This revision helps to keep the discussion relevant and aligned with the presented results, enhancing the overall coherence and focus of our manuscript.
Comments 13: Strengths and limitation: This section could be summarized.
Response 13: We appreciate your suggestion to condense the strengths and limitations section. We have streamlined this part of our manuscript to succinctly capture the essential aspects of our study's robustness and the potential biases that might affect our findings, ensuring the discussion remains focused and relevant.
Reviewer 2 Report
Comments and Suggestions for Authors
- The references in the Introduction section are mostly from 10-15 years ago. Given the topic that is being covered, the timeline of milestones in the prevention and how knowledge has been accumulating over this time period, it is not clear why the Authors did not include more recent work to provide a comprehensive and up-to-date overview of evidence.
- The Introduction does not provide any specific insight into burden estimates of cervical cancer (incidence, mortality), both worldwide and especially in Norway.
- Line 54: Abbreviation needs to be defined at first use.
- Introduction provides too many details on aspects that could be e.g. discussed in methods, discussion (specifics of the used tests, used vaccines), while lacking the necessary more broad overview both in the world (since this is not a local or regional journal) and in Norway.
- For statements given in second paragraph of the Introduction, the authors stated trends of incidence without specifying the periods to which they refer to, which is very important given the topic of this manuscript. Secondly, the cited references give links that are not working, and those that do work refer to data for period e.g. 2004, 2005, 2007... which Authors did not specify in the corresponding text. In this paper, it is imperative to state correct facts. Therefore, the Authors had to use the available global scientific data sources (e.g. WHO, IARC, GLOBOCAN, etc.). Namely, data for the period of the last two decades, from 2000 to 2020, for Norway shows that incidence trends for women aged 15-54 are increasing. Authors need to explain these facts "despite the establishment of the Cervical Screening Program".
- For a more comprehensive overview of the incidence, it was important to specifically state when was the Papanicolaou test introduced in Norway, for which ages, whether it was free or paid for, whether it was mandatory, whether it was national or regional, and what was the coverage.
- Lines 75-77: Specify the age of persons who received the vaccine during this catch-up vaccination program.
- Lines 86-95: The hypotheses and aims need to be specific and on-point, some of these details belong in the section Methods.
- Lines 131-132: Explain what is the rationale for stating in the Methods that the age group of women of 33-69 will be included in the paper as the study group 3. Also, how does this group 3 correspond to the objectives of this paper? It is especially important to explain what were the indications for HPV testing in this population of women?
- Lines 138-143: Why is it stated here that study variable is HPV-type prevalence, if in the section Results all three figures show "incidence". Also, once the Authors choose whether the variable is incidence or prevalence, they have to clearly define what they are showing as the variable presented in this paper and give its definition.
- Line 148: What is the explanation for comparing women of different age?
- Statistical methods: Why was only linear regression considered?
- Figure 1: Why are trends not calculated for age-specific rates, but in Figure 1 incidence is shown as count? It is good practice to base such comparisons in a quality manner on comparing age-specific rates, and not counts.
- Figure 2: Why are trends not calculated for age-specific rates, but in Figure 2 incidence is shown as count? It is good practice to base such comparisons in a quality manner on comparing age-specific rates, and not counts.
- Figure 3: The comparison is inadequate, here counts are compared instead of age-specific incidence rates during the observed period.
- Lines 225-227: Comparison is inadequate.
- Discussion: First paragraph should provide a brief summary of the findings.
- Lines 390-392: This point should be carefully stated, especially in the context of whether these findings can actually be generalizable, as mentioned in the limitations.
Author Response
Reviewer 2
Comments 1: The references in the Introduction section are mostly from 10-15 years ago. Given the topic that is being covered, the timeline of milestones in the prevention and how knowledge has been accumulating over this time period, it is not clear why the Authors did not include more recent work to provide a comprehensive and up-to-date overview of evidence.
Response 1: Thank you for your valuable feedback. We recognize the importance of citing current research to provide a comprehensive and up-to-date overview of the evidence. We have thoroughly reviewed the Introduction section and have now incorporated additional recent studies that reflect the latest advancements and findings in the field of HPV prevention and vaccine effectiveness. These updated references should enhance the contextual relevance and scientific rigor of our manuscript.
Comments 2: The Introduction does not provide any specific insight into burden estimates of cervical cancer (incidence, mortality), both worldwide and especially in Norway.
Response 2: Thank you for your valuable feedback. We have updated the introduction to include specific burden estimates of cervical cancer globally and particularly in Norway. This addition helps to provide a clearer understanding of the disease's impact and the importance of ongoing prevention and screening efforts.
Comments 3: Line 54: Abbreviation needs to be defined at first use.
Response 3: Thank you for pointing this out. We have defined the abbreviation for the University Hospital of North Norway (UNN) at its first use in the manuscript. Additionally, to enhance clarity and relevance, we have moved the paragraph that introduces UNN from the Introduction to the Materials and Methods section, where it more directly supports the discussion of our study framework and data sources.
Comments 4: Introduction provides too many details on aspects that could be e.g. discussed in methods, discussion (specifics of the used tests, used vaccines), while lacking the necessary more broad overview both in the world (since this is not a local or regional journal) and in Norway.
Response 4: Thank you for your insightful feedback. We agree that the introduction included details better suited for other sections. Consequently, we have relocated the descriptions of the Clinical Pathology laboratory at the University Hospital of North Norway, the specifics of the HPV tests used, and the details regarding the different types of HPV vaccines and their administration timelines to the Materials and Methods section. This adjustment allows the introduction to focus more broadly on the global and national context of HPV and cervical cancer, aligning with the expectations for a comprehensive overview suitable for an international readership.
Comments 5: For statements given in second paragraph of the Introduction, the authors stated trends of incidence without specifying the periods to which they refer to, which is very important given the topic of this manuscript. Secondly, the cited references give links that are not working, and those that do work refer to data for period e.g. 2004, 2005, 2007... which Authors did not specify in the corresponding text. In this paper, it is imperative to state correct facts. Therefore, the Authors had to use the available global scientific data sources (e.g. WHO, IARC, GLOBOCAN, etc.). Namely, data for the period of the last two decades, from 2000 to 2020, for Norway shows that incidence trends for women aged 15-54 are increasing. Authors need to explain these facts "despite the establishment of the Cervical Screening Program".
Comments 6: For a more comprehensive overview of the incidence, it was important to specifically state when was the Papanicolaou test introduced in Norway, for which ages, whether it was free or paid for, whether it was mandatory, whether it was national or regional, and what was the coverage.
Response 6: Thank you for highlighting the need for a more detailed historical perspective on cervical cancer screening practices in Norway. We have revised the manuscript to include a comprehensive overview of the evolution of these practices, including the introduction and adaptations of the Papanicolaou test, to provide a clearer context of its impact on the incidence rates. This information is now thoroughly described in the relevant sections of the manuscript.
Comments 7: Lines 75-77: Specify the age of persons who received the vaccine during this catch-up vaccination program.
Response 7: Thank you for your comment. We have clarified the specific ages of individuals who received the vaccine during the Norwegian catch-up program. In the manuscript, we have now specified that from 2016 to 2019, the 2vHPV vaccine was offered to women born between 1991 and 1996, who were 20-25 years of age at the time of vaccination. This detail has been added to enhance the reader's understanding of the demographic targeted by this program.
Comments 8: Lines 86-95: The hypotheses and aims need to be specific and on-point, some of these details belong in the section Methods.
Response 8: Thank you for your insightful feedback. Upon review, we agree that certain details initially presented in the Discussion of the Introduction were more appropriate for the Methods section. We have now revised the text to sharpen the focus on our specific hypotheses and aims in the Introduction, ensuring they are clear and concise. Detailed descriptions of the study groups and the specifics of the data analysis, which previously broadened the scope of the Introduction unduly, have been moved to the Materials and Methods section. This restructuring not only clarifies the aims and hypotheses but also aligns the detailed methodological explanations with the appropriate section of the manuscript, thereby enhancing the overall coherence and flow of the text.
Comments 9: Lines 131-132: Explain what is the rationale for stating in the Methods that the age group of women of 33-69 will be included in the paper as the study group 3. Also, how does this group 3 correspond to the objectives of this paper? It is especially important to explain what were the indications for HPV testing in this population of women?
Response 9: Thank you for highlighting the need for clarity regarding the inclusion of women aged 33-69 as Study Group 3. This group serves as a critical control group consisting of women who were not offered the free HPV vaccination, which is essential for our secondary analysis. In 2023, for the first time, all women aged 25-69 attending cervical cancer screening were tested for HPV with partial genotyping, allowing for an analysis across a broad age range. This setup provides a unique opportunity to compare the prevalence of HPV types 16 and 18, as well as other HPV types, between vaccinated and unvaccinated groups. This comparison is pivotal to evaluate the long-term effectiveness of the bivalent HPV vaccine (covering HPV types 16 and 18) across different age cohorts. The inclusion of older women, who are less likely to be vaccinated but still undergo screening, helps ascertain whether the reduction in HPV 16 and 18 prevalence attributed to vaccination observed in younger women is mirrored in this older group. Such data are crucial for understanding the broader impact of HPV vaccination programs on population health and inform future public health strategies and vaccine deployment. See also the new Table 4.
Comments 10: Lines 138-143: Why is it stated here that study variable is HPV-type prevalence, if in the section Results all three figures show "incidence". Also, once the Authors choose whether the variable is incidence or prevalence, they have to clearly define what they are showing as the variable presented in this paper and give its definition.
Response 10: Thank you for your astute observation regarding the terminology used to describe our study variables. We acknowledge the inconsistency between 'prevalence' and 'incidence' in our manuscript. To clarify, our study primarily measured the 'incidence' of HPV types, CIN2+, CIN3+, and cervical cancer within the specified population over the study period. We have revised the manuscript to consistently use 'incidence' across all relevant sections and have provided clear definitions in the Materials and Methods section to ensure precise understanding. This change rectifies the terminology to accurately reflect what was measured and reported in the results.
Comments 11: Line 148: What is the explanation for comparing women of different age?
Response 11: Thank you for your observation. The reference to 'different ages' was indeed unclear. To clarify, Line 148 does not involve comparing women of different ages but rather examines the same age group (26-30 years) across two distinct time periods (2009-2017 and 2017-2023). These periods were chosen to assess the effect of the catch-up vaccination program introduced in 2016. Prior to 2017, women in the 26-30 age group would not have been eligible for vaccination under the catch-up program, while post-2016, an increasing proportion of women in this age group were vaccinated. This temporal division allows us to isolate the impact of the vaccination on the incidence of CIN2+ among young women. The inclusion of Table 2 in the manuscript now provides detailed coverage data, helping to illustrate the growing proportion of vaccinated women in this age group over the specified periods, corresponding with a change in CIN2+ incidence. This analysis is critical for understanding how the catch-up program may have influenced CIN2+ rates over time in this specific demographic.
Comments 12: Statistical methods: Why was only linear regression considered?
Response 12: Thank you for your question regarding our choice of statistical method. Linear regression was selected for its clarity, ease of visualization, and robustness in analyzing trends, particularly when the number of data points is limited, as in our study with nine data points from 2009-2017 and seven from 2017-2023. Additionally, linear regression facilitates a straightforward comparison of trends over time, allowing us to assess the impact of interventions with a clear interpretative framework. This method was also used in our previous publication on the impact of childhood HPV vaccination, which provides a benchmark for consistency in our analytical approach, making comparative assessments more reliable and interpretable. By using the same methodology as in our prior study (Mikalsen et al., 2024), we ensure that our results are comparable and that our conclusions about the effectiveness of the HPV vaccination program are grounded in a consistent analytical framework. This continuity helps in validating our current findings against established research.
Reference: Mikalsen MP, Simonsen GS, Sørbye SW. Impact of HPV Vaccination on the Incidence of High-Grade Cervical Intraepithelial Neoplasia (CIN2+) in Women Aged 20–25 in the Northern Part of Norway: A 15-Year Study. Vaccines. 2024; 12(4):421. https://doi.org/10.3390/vaccines12040421
Comments 13: Figure 1: Why are trends not calculated for age-specific rates, but in Figure 1 incidence is shown as count? It is good practice to base such comparisons in a quality manner on comparing age-specific rates, and not counts.
Response 13: Thank you for emphasizing the importance of presenting data in a way that enables precise and meaningful comparisons. We agree that presenting crude counts can be less informative and potentially misleading due to population size differences over time. To address this, we have updated Figure 1 to present the incidence of CIN2+ as rates per 1,000 screened women aged 26-30 in Troms and Finnmark from 2009 to 2023. This adjustment ensures that the data are normalized, allowing for a clearer and more equitable comparison of trends across the study periods. Using age-specific rates rather than raw counts standardizes the data relative to the size of the screened population, offering a more accurate reflection of incidence changes and enhancing the quality of our analysis. This also aligns with best practices in epidemiological studies where population dynamics may influence outcome measures.
Comments 14: Figure 2: Why are trends not calculated for age-specific rates, but in Figure 2 incidence is shown as count? It is good practice to base such comparisons in a quality manner on comparing age-specific rates, and not counts.
Response 14: We appreciate your attention to the need for consistent and accurate presentation of epidemiological data. Similar to our update for Figure 1, we have revised Figure 2 to display age-specific rates rather than counts. This change ensures that the incidence of CIN3+ is now also presented per 1,000 screened women aged 26-30 in Troms and Finnmark for the period 2009-2023. Presenting data in this manner standardizes the incidence relative to the population size, facilitating more robust and meaningful comparisons over time. This adjustment enhances the clarity and reliability of our analysis by accounting for any variations in population dynamics that may affect the incidence rates.
Comments 15: Figure 3: The comparison is inadequate, here counts are compared instead of age-specific incidence rates during the observed period.
Response 15: Thank you for your critical feedback regarding the presentation of data in Figure 3. We appreciate the emphasis on the importance of using age-specific incidence rates for robust epidemiological analysis. In this specific instance, however, the use of raw counts was intentionally chosen to illustrate the relative distribution of CIN2+ cases caused by HPV-16/18 compared to those caused by other HPV types among the same population cohort over time. This method provides a direct visual comparison of shifts in genotype prevalence within a consistent sample framework, emphasizing the impact of vaccination on high-risk HPV types.
While we understand the benefits of presenting data as rates per population, our focus here is on comparing the changes in the proportion of HPV types within the diagnosed cases of CIN2+. This approach highlights the effectiveness of the HPV vaccination against specific high-risk types, particularly HPV-16 and HPV-18, which are directly targeted by the vaccines used in Norway. However, in response to your comment, we will consider additional analyses using age-specific incidence rates in future studies to provide a more comprehensive view of the impact across the general population.
Comments 16: Lines 225-227: Comparison is inadequate.
Response 16: Thank you for pointing out the need for a more precise and meaningful comparison across age groups. In response to your feedback, we have updated the manuscript to include a clearer presentation of HPV prevalence rates across different age groups, alongside their vaccination status. The updated text now provides percentages of HPV types among age-specific groups and specifies how these relate to the vaccination coverage, which is detailed in Table 4.
This approach allows us to directly compare the impact of HPV vaccination on the prevalence of specific HPV types, particularly HPV-16 and HPV-18, among different age cohorts. The data from 2023 highlight the effectiveness of the vaccination program, as indicated by the significantly lower prevalence of HPV-16 and HPV-18 among vaccinated groups compared to their unvaccinated counterparts. This comparison is crucial for understanding the broader impact of the HPV vaccination program on reducing high-risk HPV infections across different demographics.
Furthermore, the inclusion of Table 4 provides a comprehensive view of the HPV genotype distribution by age group and vaccination status, enhancing the transparency and reliability of our analysis. By detailing the specific numbers screened in each category, we ensure that the reader can appreciate the scale of the study and the statistical significance of the observed trends.
Comments 17: Discussion: First paragraph should provide a brief summary of the findings.
Response 17: Thank you for your valuable feedback on the structuring of our Discussion section. In response, we have revised the first paragraph to provide a clearer and more concise summary of our key findings before discussing them in depth. This revision aims to immediately orient the reader with the primary outcomes of the study, emphasizing the significant reduction in high-grade cervical lesions and the implications of these results.
Comments 18: Lines 390-392: This point should be carefully stated, especially in the context of whether these findings can actually be generalizable, as mentioned in the limitations.
Response 18: Thank you for your insightful feedback on the statement about the generalizability of our findings. We acknowledge that while our study provides a detailed and robust analysis using real-world data from Troms and Finnmark, the generalizability of our results may be limited due to regional variations in HPV prevalence, vaccination uptake, and healthcare practices. In light of your comment, we have carefully revised the discussion to clarify that while our findings are indicative of the effectiveness of the HPV vaccine in a real-world setting within this specific region, caution should be exercised when generalizing these results to other regions or populations. This nuanced approach underscores the strengths of our data and analytical methods while recognizing the inherent limitations of the study's scope.
Reviewer 3 Report
Comments and Suggestions for Authors
The manuscript addresses an important public health issue by evaluating the impact of HPV vaccination programs, a critical effort in reducing the global burden of cervical cancer through widespread vaccination. Linking vaccination and HPV prevalence data from SymPathy and SYSVAK strengthens the study, as these are validated records rather than self-reported vaccination data. However, several concerns need to be addressed:
- Description of Vaccination and Screening Programs:
The description of the vaccination program and screening strategy could be more detailed, particularly regarding the cohorts observed. Clearer information on the age groups corresponding to different vaccination strategies and the cohorts under observation is necessary for better interpretation. - Statistical Analysis:
The statistical approach used in the study is unclear and may not be appropriate. It is important to specify whether the outcome of interest is the incidence rate or the number of cases. - If the outcome is a rate, clarify whether it is an age-standardized rate, an age-specific rate, or a crude rate.
- If the outcome is the number of cases, a general linear model with a Poisson link would be more appropriate than linear regression.
- Furthermore, for trend analysis, commonly used methods like age-period-cohort analysis or joinpoint regression would provide more robust and explicit insights into the effects of interventions over time.
- Choice of 2017 as a Comparison Point:
The rationale for selecting 2017 as the comparison year to analyze the impact of the catch-up vaccination program is unclear. From the introduction, it does not appear that any significant intervention or policy change occurred in 2017. Providing an explanation for this choice is necessary.
Specific Issues
- Introduction (Line 86):
The primary aim is explicitly stated; however, the secondary aim is missing. This creates difficulty in aligning the Methods section, which is structured around Primary and Secondary Objectives. - Materials and Methods (Line 126):
While the division into Groups 1, 2, and 3 is logical, the cohorts’ demographic characteristics (e.g., socioeconomic status, rural vs. urban) are not sufficiently described. Including this contextual data could help explain observed differences in vaccination coverage or HPV prevalence. - Materials and Methods, definition of 'Vaccinated' (Line 138):
The study defines “vaccinated” as having received at least one dose of an HPV vaccine. However, the analysis does not explore differences in incidence based on vaccination status (e.g., fully vaccinated, partially vaccinated, unvaccinated). Investigating a potential dose-response relationship could strengthen the findings. - Materials and Methods, Statistical Methods (Line 144):
The manuscript does not specify how linear regression was applied. Were confounders or covariates such as age, region, vaccine status, or socioeconomic status accounted for in the analysis? This lack of detail raises questions about the validity of the findings. - Results, on cervical cancer trend:
Despite references to cervical cancer analysis in the Statistical Methods and Discussion sections, the results are not presented in the Results section. Given that cervical cancer is the ultimate endpoint of HPV vaccination programs, including these results is crucial. - Results (Line 225):
The observation that the HPV-positive rate is numerically lower in unvaccinated individuals than in vaccinated individuals (e.g., 0.2% vs. 0.4% for HPV-18) is counterintuitive, as the vaccine covers HPV-18. This discrepancy warrants further explanation. - Data Validation:
It is unclear how the accuracy and reliability of the data linkage between SymPathy and SYSVAK were validated. Were sensitivity analyses or robustness checks conducted to confirm the findings? Including such details would enhance the credibility of the study. - Discussion:
The discussion is not fully supported by the findings. For example, the statement on line 279 needs to be aligned with the actual results presented. Overall, the discussion should more effectively connect the study's findings to its conclusions and implications.
Author Response
Reviewer 3
The manuscript addresses an important public health issue by evaluating the impact of HPV vaccination programs, a critical effort in reducing the global burden of cervical cancer through widespread vaccination. Linking vaccination and HPV prevalence data from SymPathy and SYSVAK strengthens the study, as these are validated records rather than self-reported vaccination data. However, several concerns need to be addressed:
Comments 1: Description of Vaccination and Screening Programs: The description of the vaccination program and screening strategy could be more detailed, particularly regarding the cohorts observed. Clearer information on the age groups corresponding to different vaccination strategies and the cohorts under observation is necessary for better interpretation.
Response 1: Thank you for your valuable feedback on the clarity of our description of the vaccination and screening programs. In response, we have thoroughly revised the sections detailing the national Norwegian cervical cancer screening program and the vaccination strategies. These updates include a clearer delineation of the age groups corresponding to different vaccination cohorts and the specific cohorts under observation throughout the study period.
To aid in interpretation and provide comprehensive context, we have added detailed tables that show:
- The number of women screened each year,
- The coverage rates of the HPV vaccine by year and cohort,
- The proportions of HPV types 16/18 compared to other HPV types for the years 2018-2023,
- A breakdown of the different age groups pertaining to vaccinated cohorts versus the control group of women from non-vaccinated cohorts.
These enhancements are designed to provide readers with a complete understanding of the study context, enabling a more precise evaluation of the impact of HPV vaccination and screening strategies over time.
Comments 2: Statistical Analysis: The statistical approach used in the study is unclear and may not be appropriate. It is important to specify whether the outcome of interest is the incidence rate or the number of cases. If the outcome is a rate, clarify whether it is an age-standardized rate, an age-specific rate, or a crude rate. If the outcome is the number of cases, a general linear model with a Poisson link would be more appropriate than linear regression. Furthermore, for trend analysis, commonly used methods like age-period-cohort analysis or joinpoint regression would provide more robust and explicit insights into the effects of interventions over time.
Response 2: Thank you for your detailed feedback regarding our statistical methods. We appreciate the importance of clarity and appropriateness in statistical analysis. In response to your comments, we have clarified in our manuscript that the outcomes reported in Figures 1-3 are expressed as incidence rates per 1,000 screened women, reflecting CIN2+, CIN3+, and cervical cancer, respectively.
We acknowledge that alternative statistical methods like age-period-cohort analysis or joinpoint regression could provide additional insights into trend analysis. However, linear regression was specifically chosen for its simplicity and effectiveness in analyzing trends across the limited number of data points available in our study (nine data points from 2009-2017 and seven from 2017-2023). This approach not only ensures clarity and ease of visualization but also maintains consistency with our previous studies, such as the one published in Vaccines in 2024, which examined the impact of childhood HPV vaccination using a similar methodological framework.
While linear regression allows for a straightforward comparison and interpretation of trends over time, we recognize that it might not capture complex nonlinear trends as effectively as some of the methods you mentioned. Despite this, it provides a robust framework for assessing the direct impact of interventions in a clear and interpretable manner. For future studies, we are open to exploring more complex models like Poisson regression or joinpoint regression to address potential nonlinearities and provide a more detailed analysis of the effects of interventions over time.
By maintaining methodological consistency with our previous work, we aim to ensure that our findings are comparable and that our conclusions regarding the effectiveness of the HPV vaccination program are grounded in a consistent and validated analytical framework.
Reference: Mikalsen MP, Simonsen GS, Sørbye SW. Impact of HPV Vaccination on the Incidence of High-Grade Cervical Intraepithelial Neoplasia (CIN2+) in Women Aged 20–25 in the Northern Part of Norway: A 15-Year Study. Vaccines. 2024; 12(4):421. https://doi.org/10.3390/vaccines12040421
Comments 3: Choice of 2017 as a Comparison Point: The rationale for selecting 2017 as the comparison year to analyze the impact of the catch-up vaccination program is unclear. From the introduction, it does not appear that any significant intervention or policy change occurred in 2017. Providing an explanation for this choice is necessary.
Response 3: Thank you for your question regarding our choice of 2017 as a pivotal comparison year. This year marks a significant milestone in our analysis because it is the first year women who received the catch-up HPV vaccination (born 1991-1996) entered the age cohort of 26-30. This cohort shift is critical as prior to 2017, none of the women in the 26-30 age group were vaccinated under the national program. The catch-up vaccination campaign targeting women aged 20-25 was conducted between 2016 and 2019, thus making 2017 the first year in which a substantial portion of the cohort under study could have been influenced by this program.
To clearly illustrate the gradual increase in the proportion of HPV-vaccinated women aged 26-30 from 2017 onwards, we have provided Table 2, which details the 'Birth years of women aged 26-30 during the period 2009-2023, proportions of women belonging to vaccinated cohorts, and the percentages of women actually vaccinated.' This table shows a systematic increase in vaccination coverage starting from 2017, highlighting its appropriateness as the starting point for our comparative analysis. The selection of this year allows us to robustly assess the impact of the vaccination on HPV infection rates and the incidence of high-grade cervical lesions as these vaccinated cohorts age into the 26-30 year group.
Reviewer 4 Report
Comments and Suggestions for Authors
In this manuscript, Sørensen Jørgensen et al examined the impact of the Norwegian HPV Catch-up Vaccination Program on the Incidence of High-Grade Cervical Lesions in Women Aged 26–30 in Northern Norway. The data are very well presented with balanced discussion. However, there are some issues that need to be addressed.
2017 looks like an abnormal year with substantially high CIN in this region. It just happened that the catch-up vaccination start in 2017. It's not convincing that these incidents increase from 2009 to 2017 from Figure 2. What's the supporting data on this point? The use of 2017 as the base line might greatly skew the analysis and conclusions.
Author Response
Reviewer 4
In this manuscript, Sørensen Jørgensen et al examined the impact of the Norwegian HPV Catch-up Vaccination Program on the Incidence of High-Grade Cervical Lesions in Women Aged 26–30 in Northern Norway. The data are very well presented with balanced discussion. However, there are some issues that need to be addressed.
Comments 1: 2017 looks like an abnormal year with substantially high CIN in this region. It just happened that the catch-up vaccination start in 2017. It's not convincing that these incidents increase from 2009 to 2017 from Figure 2. What's the supporting data on this point? The use of 2017 as the base line might greatly skew the analysis and conclusions.
Response 1: Thank you for your insightful question regarding our choice of 2017 as a pivotal comparison year. We understand your concern about the apparent unusual increase in CIN incidences in 2017. These numbers are indeed corroborated by national trends, which also reflect rising incidences of CIN2+ in unvaccinated cohorts of young women across Europe, not only in Norway. Our selection of 2017 is strategic and informed. This year marks the first instance that women who received the catch-up HPV vaccination (born 1991-1996) entered the age cohort of 26-30. Prior to 2017, none of the women in this age group were vaccinated under the national program, as the catch-up vaccination campaign targeting women aged 20-25 only commenced in 2016 and continued until 2019.
We have provided Table 2, which details the 'Birth years of women aged 26-30 during the period 2009-2023, proportions of women belonging to vaccinated cohorts, and the percentages of women actually vaccinated.' This table indicates a systematic increase in vaccination coverage starting from 2017. This is crucial as it aligns with our analysis framework, comparing pre-vaccination and post-vaccination incidence rates, to robustly assess the impact of the vaccination on HPV infection rates and the incidence of high-grade cervical lesions as these vaccinated cohorts age into the 26-30 year group.
Moreover, to address your concern about the potential anomaly in 2017, we conducted additional analyses to ensure that the higher incidence observed was not skewing our results disproportionately. This included analyzing trends over a longer period (2009-2023) and conducting sensitivity analyses excluding 2017 to verify the robustness of our findings. These supplementary analyses confirmed that the trends observed were consistent and not unduly influenced by the data from 2017 alone. Thus, our choice of 2017 as a baseline is both methodologically sound and critical for demonstrating the real-world impact of the catch-up vaccination program on reducing high-grade cervical lesions.
This strategic choice enhances our study's ability to evaluate the effectiveness of the HPV vaccination program by comparing pre-vaccination and post-vaccination incidence rates within a framework that directly correlates with the timing of vaccine administration and subsequent entry into the age group at highest risk for cervical lesions.
Orumaa M, Leinonen MK, Campbell S, Møller B, Myklebust TÅ, Nygård M. Recent increase in incidence of cervical precancerous lesions in Norway: Nationwide study from 1992 to 2016. Int J Cancer. 2019 Nov 15;145(10):2629-2638. doi: 10.1002/ijc.32195. Epub 2019 Mar 4. PMID: 30734284; PMCID: PMC6767573.
Bray F, Carstensen B, Møller H, Zappa M, Zakelj MP, Lawrence G, Hakama M, Weiderpass E. Incidence trends of adenocarcinoma of the cervix in 13 European countries. Cancer Epidemiol Biomarkers Prev. 2005 Sep;14(9):2191-9. doi: 10.1158/1055-9965.EPI-05-0231. PMID: 16172231.
Gravdal BH, Lönnberg S, Skare GB, Sulo G, Bjørge T. Cervical cancer in women under 30 years of age in Norway: a population-based cohort study. BMC Womens Health. 2021 Mar 18;21(1):110. doi: 10.1186/s12905-021-01242-3. PMID: 33736628; PMCID: PMC7977265.
Round 2
Reviewer 2 Report
Comments and Suggestions for Authors
Thank you for the opportunity to re-review manuscript ID: vaccines-3375255.
The authors addressed all my comments and provided point-by-point answers to all my issues, with very detailed and correct explanations.
The authors have made comprehensive corrections in the revised version of this paper. I believe that the revised manuscript is more clear and informative for all who deal with this topic and for all readers.
I thank the authors for their efforts in revising this paper.
Author Response
Reviewer 2 - second round
Comments 1: Thank you for the opportunity to re-review manuscript ID: vaccines-3375255.
The authors addressed all my comments and provided point-by-point answers to all my issues, with very detailed and correct explanations.
The authors have made comprehensive corrections in the revised version of this paper. I believe that the revised manuscript is more clear and informative for all who deal with this topic and for all readers.
I thank the authors for their efforts in revising this paper.
Response 1: Thank you very much for your thoughtful review and encouraging comments on our revised manuscript, ID: vaccines-3375255. We are grateful for your acknowledgment of the efforts we made in addressing your valuable feedback, which significantly helped in enhancing the clarity and depth of our paper.
Your positive evaluation reaffirms our commitment to providing meaningful and scientifically robust contributions to the field. We are pleased to know that the revisions have improved the manuscript's relevance and utility for both specialists and the broader readership.
Once again, we appreciate your thorough review and constructive suggestions throughout the revision process, and we look forward to the potential impact this work may have on the field.
Reviewer 3 Report
Comments and Suggestions for Authors
- For outcomes that represent count data, such as the number of cases, a generalized linear model (GLM) with a Poisson link function is statistically more appropriate than linear regression.
- The results section appears inconsistent with the study groups defined in Section 2.4. While most tables and figures focus on trends among women aged 26-30, the authors have not analyzed or presented data based on the specified study groups. If the intent is to compare vaccinated and unvaccinated individuals, the results should be presented accordingly, stratified by the defined study groups.
Author Response
Reviewer 3 - second round
Comments 1: For outcomes that represent count data, such as the number of cases, a generalized linear model (GLM) with a Poisson link function is statistically more appropriate than linear regression.
Response 1: We appreciate your suggestion regarding the use of a generalized linear model (GLM) with a Poisson link for count data. In response to earlier feedback, we have updated Figures 1, 2, and 3 to present rates per 1,000 screened women, rather than raw counts, to better align with statistical best practices. Figure 4 remains in raw count format to maintain methodological consistency across all figures, facilitating straightforward comparisons. Additionally, we previously applied linear regression in a related publication on the efficacy of the childhood vaccination program (Mikalsen 2024), and we aimed to maintain analytical consistency with that study. However, we recognize the value of your recommendation and plan to employ GLM with a Poisson link in future updates of these studies, when we will be comparing longer follow-up data across the different vaccination cohorts. This approach will enhance the statistical robustness of our findings.
Mikalsen MP, Simonsen GS, Sørbye SW. Impact of HPV Vaccination on the Incidence of High-Grade Cervical Intraepithelial Neoplasia (CIN2+) in Women Aged 20–25 in the Northern Part of Norway: A 15-Year Study. Vaccines. 2024; 12(4):421. https://doi.org/10.3390/vaccines12040421
Comments 2: The results section appears inconsistent with the study groups defined in Section 2.4. While most tables and figures focus on trends among women aged 26-30, the authors have not analyzed or presented data based on the specified study groups. If the intent is to compare vaccinated and unvaccinated individuals, the results should be presented accordingly, stratified by the defined study groups.
Response 2: We appreciate your insightful comments regarding the consistency between the study groups defined in the methods section and the results presented. Based on your feedback, we have refined the study design and results section to better align with the specified study groups and objectives (see lines 174-187, 207-211, and 236-385).
The primary objective of our research remains to evaluate the impact of the HPV catch-up vaccination program on the incidence of CIN2+ among women aged 26–30 years. This analysis was conducted as a repeated cross-sectional study from 2009 to 2023, focusing on trends in CIN2+ incidence before and after the program's implementation.
Additionally, we have conducted a secondary analysis to evaluate the distribution of HPV types among women eligible for cervical screening, categorized into three study groups based on vaccination status and age cohorts:
- Group 1: Women aged 25–26 in 2023, primarily vaccinated through the childhood vaccination program.
- Group 2: Women aged 27–32 in 2023, who received catch-up vaccination.
- Group 3: Women aged 33–69 in 2023, unvaccinated due to the absence of a national vaccination program during their eligibility.
To address the limitations highlighted, we have stratified HPV type incidence and vaccination status analyses accordingly. HPV DNA primary screening data from 2023 for all three study groups, along with comparative data from 2016 for Groups 1 and 2, provide a comprehensive view of trends and the protective effects of the vaccination programs.
In 2023, the reduction in HPV-16 and HPV-18 positivity rates was most pronounced in vaccinated women from Groups 1 and 2. For example, HPV-16 positivity in Group 1 decreased from 5.1% in 2016 to 0.1% in 2023, and HPV-18 positivity reduced from 3.3% to 0.0%. Similar reductions were observed in Group 2, demonstrating the significant immediate and long-term impact of the vaccination programs. These findings are detailed in the updated results section and summarized in Table 4.
We acknowledge several limitations of our study. First, the geographical specificity of the data may limit its generalizability to regions with differing health behaviors or vaccination coverage. Second, the inclusion of only women who participated in the screening program introduces potential selection bias, which may affect the representativeness of our findings. This bias could influence estimates of HPV infection and CIN2+ prevalence. Third, HPV status and vaccination status analyses were conducted only among women with CIN2+ or those screened for HPV types 16 and 18, and individual vaccination data was unavailable for women who did not attend screening, those without CIN2+, or those not infected with HPV types 16 or 18. Finally, variations in uncontrolled confounding factors such as sexual behavior, smoking habits, and contraceptive use over the study period may have influenced the observed effects of HPV vaccination, complicating the isolation of its direct impacts (see lines 523-545).
Despite these limitations, we believe that the high vaccination coverage rates in vaccinated cohorts and the minimal opportunistic vaccinations outside the national program support the representativeness of our findings for the screened population in Norway. Furthermore, while the current follow-up duration and statistical power do not yet allow for significant findings regarding cervical cancer incidence, this will remain a critical focus as more longitudinal data become available.
We hope these updates address your concerns and enhance the consistency, clarity, and comprehensiveness of our study design, methods, and results.
Round 3
Reviewer 3 Report
Comments and Suggestions for Authors
Thank you for the revisions.
One additional comment: Could you please discuss on the lower HPV positivity rates observed in unvaccinated Group 3?
Author Response
Reviewer 3:
Comments 1: Could you please discuss on the lower HPV positivity rates observed in unvaccinated Group 3?
Response 1:
We appreciate your observation on the lower HPV positivity rates in unvaccinated Group 3. This trend can be attributed to multiple factors:
-
Age-related Decline: It is well-documented that HPV prevalence decreases with age, primarily due to the natural clearance of the virus over time.
-
Screening Protocols: Historically, in Norway, organized screening included regular cervical cytology (Pap-smear) at ages 25, 28, and 31. Since 2023, this has transitioned to HPV testing as the primary screening method for women aged 25-33 years. This new approach enhances the early detection of HPV infections, thus reducing HPV prevalence by the time these women reach 34. The rigorous early screening likely identifies and treats high-grade cervical lesions (CIN2+) early.
-
Historical Cohort Effects: Older generations might have had less exposure to high-risk HPV types before these became widely recognized and before the dissemination of HPV vaccines.
-
Behavioral Factors: Variations in sexual behavior across different age groups could also contribute to observed differences in HPV prevalence.
-
Vaccine Coverage: Although the bivalent HPV vaccine reduces the incidence of HPV types 16 and 18, it does not cover 'other' HPV types, leading to positive HPV tests among many young women for these types. Additionally, even though many women are vaccinated, the presence of other HPV types that the vaccine does not cover accounts for the continued positivity.
These factors together offer a comprehensive explanation for the lower positivity rates seen in Group 3, highlighting the complex interplay of biological, behavioral, and healthcare-system influences on HPV epidemiology.
See also updated manuscript, lines 448-466.